# Beyond Shared Vocabulary: Increasing Representational Word Similarities across Languages for Multilingual Machine Translation

**Di Wu     Christof Monz**
Language Technology Lab
University of Amsterdam
{d.wu, c.monz}@uva.nl

## Abstract

Using a vocabulary that is shared across languages is common practice in Multilingual Neural Machine Translation (MNMT). In addition to its simple design, shared tokens play an important role in positive knowledge transfer, assuming that shared tokens refer to similar meanings across languages. However, when word overlap is small, especially due to different writing systems, transfer is inhibited. In this paper, we define word-level information transfer pathways via word equivalence classes and rely on graph networks to fuse word embeddings across languages. Our experiments demonstrate the advantages of our approach: 1) embeddings of words with similar meanings are better aligned across languages, 2) our method achieves consistent BLEU improvements of up to 2.3 points for high- and low-resource MNMT, and 3) less than 1.0% additional trainable parameters are required with a limited increase in computational costs, while inference time remains identical to the baseline. We release the codebase to the community.[1]

## 1 Introduction

Multilingual systems (Johnson et al., 2017; Lample and Conneau, 2019) typically use a shared vocabulary to build the word space uniformly. For instance, in MNMT scenarios, this is achieved by combining all source and target training sentences together and training a shared language-agnostic tokenizer, e.g., BPE (Sennrich et al., 2016), to split the words into tokens.

Such a design is simple and scales easily. Moreover, shared tokens also encourage positive knowledge transfer when they refer to equivalent or similar meanings. Research on cross-lingual word embeddings (Søgaard et al., 2018; Ruder et al., 2019) shows that exploiting a weak supervision signal from identical words remarkably boosts cross-

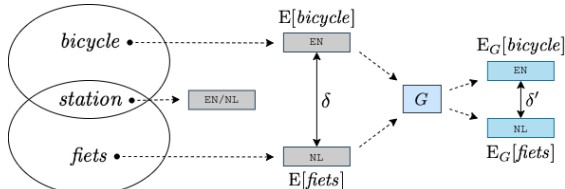

Figure 1: *bicycle* and *fiets* have the same meaning, but use different forms, potentially leading to a larger distance $\delta$ between their embeddings (E[·]). Our graph-based module $G$ explicitly reparameterizes the word embeddings ($E_G$[·]) leading to a reduced distance $\delta'$.

lingual word embedding or bilingual lexicon induction, i.e., word translation. This point is also held for higher-level representation learning, e.g., multilingual BERT (Devlin et al., 2019), where Pires et al. (2019) show that knowledge transfer is more pronounced between languages with higher lexical overlap. For bilingual machine translation, Aji et al. (2020) investigate knowledge transfer in a parent-child transfer setting[2] and reveal that word embeddings play an important role, particularly if they are correctly aligned.

All findings above point to the importance of word-level knowledge transfer for downstream systems, no matter whether this transfer is achieved through sharing, mapping, or alignment. To some extent, knowledge transfer in multilingual translation systems (Johnson et al., 2017) can be seen as a special case in the aforementioned bilingual translation setting, where model parameters and vocabulary are both shared and trained from scratch, and then knowledge transfer should emerge naturally. However, the shared vocabulary, as one of the core designs, has limitations: 1) When languages use different writing systems, there is little word overlap and knowledge sharing suffers. 2) Even if languages use similar writing systems, shared tokens may have completely different meanings in

---

[2]Normally, it refers to tuning the child model on the pre-trained parent model, while the embedding table could be fully shared, partially shared, or not shared at all.

different languages, increasing ambiguity.

In this paper, we target the first issue of word-level knowledge sharing across languages. As illustrated in Figure 1, due to different surface forms, *bicycle* in English and *fiets* in Dutch have the same meaning but are placed differently in the embedding space, unlike the shared word *station*. Our goal is to pull together embeddings of meaning-equivalent words in different languages, including languages with different writing systems, facilitating knowledge transfer for downstream systems. For simplicity, we choose English as the pivot and encourage words in other languages to move closer to their English counterparts.

To this end, we define and mine subword-level knowledge propagation paths and integrate them into a graph which is represented as an adjacency matrix. Then, we leverage graph networks (Welling and Kipf, 2016) to model information flow. The embeddings of meaning-equivalent words will transfer information along with the paths defined in the graph and finally converge into a re-parameterized embedding table, which is used by the MNMT model as usual. At a higher level, our approach establishes priors of word equivalence and then injects them into the embedding table to explicitly encourage knowledge transfer between words in the vocabulary.

We choose multilingual translation as test bed to investigate the impact of word-based knowledge transfer. Several experiments show the advantages of our approach: 1) Our re-parameterized embedding table exhibits enhanced multilingual capabilities, resulting in consistently improved alignment of word-level semantics across languages, encouraging word-level knowledge transfer beyond identical surface forms. 2) Our method consistently outperforms the baseline by a substantial margin (up to 2.3 average BLEU points) for high- and low-resource MNMT, which empirically demonstrates the benefits of our re-parameterized embeddings. 3) Our method scales: The extra training time and memory costs are small, while the inference time is exactly the same as benchmark systems. Moreover, we demonstrate our method adapts to massive language pairs and a large vocabulary.

## 2 Related Work

Prior work has demonstrated the importance of word-level knowledge transfer. For instance, Gouws and Søgaard (2015) show that by replacing some specific words with their cross-lingual equivalent counterparts, a downstream cross-lingual part-of-speech tagging task can benefit from its high-resource counterpart. Using cross-lingual word embedding (Søgaard et al., 2018; Ruder et al., 2019) can be seen as an extension of this idea at the representation level. E.g., by fixing and applying pre-trained embeddings to cross-lingual tasks.

For transfer learning in bilingual NMT, Amrhein and Sennrich (2020) show that the child and parent models should share a substantial part of the vocabulary. For languages with different scripts, vocabulary overlap is minimal and transfer suffers. To alleviate this issue, they leverage a romanization tool, which maps characters in various scripts to Latin characters, often based on pronunciation, to increase overlap. However, romanization has some limitations because 1) romanization operates at the character level, sometimes including pronunciation features, and hence its benefits are mostly limited to named entities, and 2) the process of romanization is not always reversible.

Sun et al. (2022) extend surface-level characters normalization and apply it to multilingual translation. Besides romanization signals, they further introduce phonetic and transliterated signals to augment training data and unify writing systems. Their approach also shows the merits of larger vocabulary overlaps. However, the aforementioned limitations of romanization and transliteration still hold.

Compared to previous work, our method is more general and practical. We mine equivalent words at the semantic level and inject priors into embeddings to improve knowledge transfer. Firstly, it avoids the flaws in romanization- or pronunciation-based approaches. Secondly, the graph-based transfer framework naturally adapts to multilingual scenarios, where many-to-many equivalences exist.

## 3 Reparameterization Framework

In this section, we describe 1) how to build an equivalence graph using pair-wise data and 2) how to re-parameterize the embeddings via our graph-based approach. Note that in the discussion below we use the term 'word' as referring to an actual word or a subword if a word segmentation approach such as BPE (Sennrich et al., 2016) is used.

### 3.1 Equivalence Graph Building

We define the words with the same meaning in different languages as equivalent words classes,

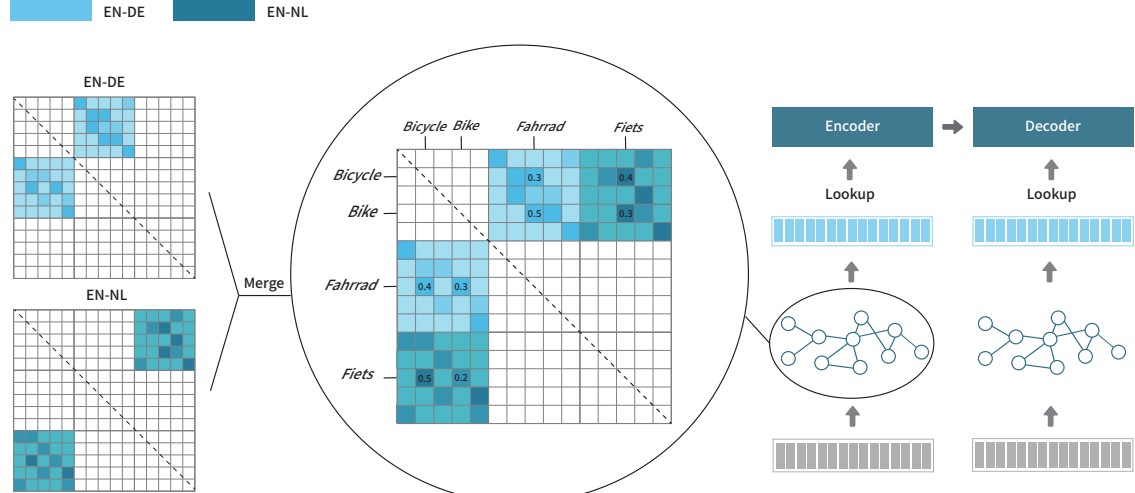

Figure 2: Illustration of our framework. The left part denotes the subgraphs we build for each language pair, e.g., EN-DE and EN-NL, which are further merged into a multilingual graph. Since we only rely on English-centric data, the graph is sparse, and only four (of the nine possible) sub-matrices are filled. As shown in the right part, the information from the original embeddings (in grey) transfer and converge into the re-parameterized embeddings (in blue) along the pathways defined in the graph, which are further used by a standard encoder-decoder model. All parameters, including embeddings, are trained from scratch.

e.g., {*bicycle*, *bike*, *Fahrrad*, *fiets*} all mean bike and therefore belong to the same equivalence class. This semantic equivalence involves many-to-many relationships across languages, which we represent in a graph as follows. For these words, we aim to transfer information with each other through the graph and make them converge into similar embedding representations.

Given a shared vocabulary $V$ with size $|V|$, we define the corresponding equivalence graph as an adjacency matrix $G^{|V| \times |V|}$. Each point $g_{i,j} \in G$ is a non-negative real number between 0.0 to 1.0 and denotes the ratio of information transferred from the word $v_j$ to word $v_i$ in $V$. In our approach, we mine equivalent words class via word alignment and define the corresponding transfer ratios base on them.

More formally, let $D$ be the entire (multilingual) bitext corpus, $D_{s,t} \subseteq D$ denotes a subset of parallel data with a translation direction from source language $s$ to target language $t$. We train and extract all of the subword-level alignments for each $D_{s,t}$ separately. For two words $v_i$ and $v_j$ in $V$, we define an alignment probability from $v_j$ to $v_i$ in $D_{s,t}$ as corresponding transfer ratios $g_{i,j}^{s,t}$ as follows:

$$g_{i,j}^{s,t} = \frac{c_{i,j}^{s,t}}{\sum_{k=1}^{|V|} c_{i,k}^{s,t}}, \tag{1}$$

where $c_{i,j}^{s,t}$ is the number of times both words are aligned with each other across $D_{s,t}$.

A higher ratio is derived from a greater pairwise occurrence in the bitext. This is based on the intuition that when a pair of aligned words frequently co-occur, they 1) have higher confidence as equivalent words, and 2) the knowledge sharing between these two will benefit more context during training. The corresponding bilingual equivalence graph $G^{s,t}$ can be induced by filling an adjacency matrix using $g_{i,j}^{s,t}$. By element-wise summation of multiple $G^{s,t}$, we can merge multiple bilingual graphs into a single multilingual graph $G$:

$$G = \sum_{D_{s,t} \subseteq D} G^{s,t}, \tag{2}$$

and further normalize $G$ to guarantee that each row sums to 1.0.

Figure 2 illustrates our equivalence graph $G$ for two language pairs, i.e., EN-DE and EN-NL. It is worth noting that, due to only English-centric alignments being extracted, the spaces of non-English direction are usually empty (such as German-Dutch).

For practicality, in this paper, we choose English as the pivot and define transfer paths based on alignments between English and other languages. It is worth mentioning that alternative approaches, like leveraging multilingual vocabulary (Lample et al., 2018b) or multilingual colexification graphs (Liu et al., 2023), could be used as well.

## 3.2 GNN based Messaging Passing

In Section 3.1, we model the priors of word equivalences as a graph. Here, we show how to inject such priors into embeddings via graph networks, thereby re-parameterizing the embedding table.

The graph can be represented as an adjacency matrix $G^{|V| \times |V|}$, see Section 3.1. Let $E \in \mathbb{R}^{|V| \times d}$ be the original embedding table containing all $|V|$ original word embeddings, where $d$ is the representation dimensionality.

Given $G$ and $E$, we can easily define various information transfers on the graph as matrix operations. E.g., a weighted sum of meaning-equivalent word embeddings can be defined as $E' = G E$.

Graph networks (Welling and Kipf, 2016) extend such operations in a multilayer neural network fashion. In each layer, non-linear functions and learnable linear projections are also involved to adjust the aggregation of messages. Here, the re-parameterized word embeddings derived from the first-layer graph network are defined as follows:

$$\mathbf{e}'_i = \rho(W_1 \mathbf{e}_i + W_2 \sum_{j \in \mathcal{N}(i)} g_{i,j} \cdot \mathbf{e}_j + \mathbf{b}), \quad (3)$$

where $\mathbf{e}_i = E[v_i]$, i.e., the embedding of word $v_i$, and $W_1, W_2 \in \mathbb{R}^{d \times d}$ and $\mathbf{b} \in \mathbb{R}^d$ are learnable parameters, and $\rho$ is a non-linear activation function, such as ReLu (Glorot et al., 2011). $\mathcal{N}(i)$ denotes a set of neighbors of $i$-th node in the graph, i.e., the aligned words with $g_{i,\cdot} > 0$. Respectively, $W_1$ learns the projection for the current word embedding $\mathbf{e}_i$ and $W_2$ learns for each neighbors $\mathbf{e}_{j \in \mathcal{N}(i)}$.

Equivalently, we can rewrite Equation 3 in a matrix fashion as follows:

$$E' = \rho(EW_1 + GEW_2 + B). \quad (4)$$

To allow the message to pass over multiple hops, we stack multiple graph networks and calculate representations recursively as follows:

$$E^{h+1} = \rho(E^h W_1^h + GE^h W_2^h + B^h), \quad (5)$$

where $h$ is the layer index, i.e., *hop*, and $E^0$ is equal to the original embedding table $E$. The last layer representation $E^H$ is the final re-parameterized embedding table, for the maximum number of hops $H$, which is then used by the system just like any vanilla embedding table.

Figure 2 illustrates the overall architecture. The information from the original embedding table propagates through multi-hop graph networks and

converges to the re-parameterized table. Then, the downstream standard MNMT system looks up corresponding word embeddings from the re-parameterized table. The whole architecture is end-to-end and supervised by the translation objective.

It is worth noting that, although the graph we build in Section 3.1 only contains English-centric pathways, knowledge transfer beyond English-centric directions can also be handled through the multi-hop mechanism. E.g., words in Dutch will transfer information to English counterparts first and then further propagate to other languages, e.g., German, by a 2-hop mechanism. We empirically evaluate this point in Section 5.2.

## 4 Experiments and Results

In this section, we apply our approach to train multilingual translation models under different configurations.

### 4.1 Experimental Setup

#### 4.1.1 Datasets

We conduct experiments on two datasets, the smaller IWSLT14 benchmark and our own large-scale dataset called EC30.

For IWSLT14, we follow the setting of Lin et al. (2021) and collect 8 English-centric language pairs, with size ranging from 89k to 169k. The data processing script[3] follows Tan et al. (2018).

To ensure a more diverse and inclusive large-scale evaluation, we used the EC30 dataset derived from EC40 (Tan and Monz, 2023), where we excluded the data of 10 super low-resource languages. The EC30 dataset consists of 61 million English-centric bilingual sentences as training data, covering 30 non-English languages across a wide spectrum of resource availability, ranging from High (5M) to Medium (1M), and Low (100K) resources. Each resource group consists of languages from 5 families with multiple writing systems. We choose Ntrex (Federmann et al., 2022) and Flores-101 (Goyal et al., 2022) as our validation and test datasets, respectively. A more detailed description of datasets is provided in Appendix A.1.

We tokenize data with Moses (Koehn et al., 2007) and use SentencePiece[4] with BPE (Sennrich et al., 2016) with 30K and 128K merge operations

---

[3]https://github.com/RayeRen/
multilingual-kd-pytorch/blob/master/data/iwslt/
raw/prepare-iwslt14.sh
[4]https://github.com/google/sentencepiece

| Model | DE | ES | FA | AR | HE | NL | PL | IT | EN→X | X→EN | AVG |
|---|---|---|---|---|---|---|---|---|---|---|---|
| Baseline (Lin et al., 2021) | 28.1 | 35.2 | 16.9 | 20.9 | 29.0 | 30.9 | 16.4 | 29.2 | - | - | 25.8 |
| LASS (Lin et al., 2021) | 29.8 | 37.3 | 17.9 | 22.9 | 30.9 | 33.0 | 17.9 | 30.9 | - | - | 27.6 |
| Our Baseline | 28.5 | 36.0 | 17.4 | 20.2 | 27.9 | 31.5 | 17.6 | 29.7 | 24.4 | 27.8 | 26.1 |
| Weighted Sum | 29.2 | 36.7 | 18.1 | 20.9 | 28.5 | 32.2 | 18.2 | 30.5 | 24.8 | 28.7 | 26.8 |
| GraphMerge-1hop | 30.2 | 37.5 | 19.0 | 21.7 | 30.0 | 33.4 | 18.8 | 31.3 | 25.4 | 30.0 | 27.7 |
| GraphMerge-2hop | 30.4 | 37.9 | 19.0 | 21.9 | 30.0 | 33.7 | 19.2 | 31.6 | **25.5** | 30.5 | 28.0 |
| GraphMerge-3hop | **30.7** | **38.2** | **19.9** | **22.3** | **30.1** | **34.0** | **19.4** | **32.2** | 25.4 | **31.3** | **28.4** |
| 3-hop Gain | +2.2 | +2.2 | +2.5 | +2.1 | +2.2 | +2.5 | +1.8 | +2.5 | +1.0 | +3.5 | +2.3 |

Table 1: Results on the IWSLT14 dataset. Following previous work (Lin et al., 2021), we report average out-of- and into-English BLEU scores. For instance, the numbers on the DE column are the average of EN→DE and DE→EN BLEU scores. EN→X and X→EN denote the average performance on 8 language pairs. We show the results from Lin et al. (2021) in the first block, which learns language-specific sub-network for MNMT. Also, we report our reproduced baseline results. *3-hop Gain* are the gains over the reproduced baseline. The best results in each column are in bold. More detailed results can be found in Appendix A.3.

for IWSLT14 and EC30, respectively. For EC30, we employ temperature sampling to select data to train BPE. The temperature is aligned with that of the MNMT training phase.

### 4.1.2 Training Settings

For IWSLT14, we follow the setting of Lin et al. (2021), using a standard 6-layer encoder 6-layer decoder transformer model with 4 attention heads, 512 embedding dimensions, and 1,024 feedforward dimensions. For EC30, we use Transformer-Big with 16 attention heads, 1,024 embedding dimensions, and 4,096 feedforward dimensions. All models are trained in a many-to-many setting.

The learning rate is 5e-4 with 4,000 warmup steps and a *inverse sqrt* decay schedule. All dropout rates and label smoothing are set to 0.1. Data from different language pairs are sampled with a temperature of 2.0 and 5.0 for IWSLT14 and EC30, respectively. We train all models with an early-stopping strategy[5] and evaluate by using the best checkpoint as selected based on the loss on the development set. More detailed training settings can be found in Appendix A.2

Tokenized BLEU is used as the metric in the content. To show a consistent improvement across metrics, SacreBLEU (Post, 2018)[6], ChrF++ (Popović, 2017), and Comet (Rei et al., 2020) are also reported in Appendix A.3 and Appendix A.4.

If not mentioned otherwise, we use *eflomal*[7] with *intersect* alignment symmetrization to extract alignments and build graphs as described in Section 3.1.

### 4.2 Results on IWSLT14

Table 1 summarizes the results for IWSLT14. The *Weighted Sum* represents our most naïve setting, i.e., the left multiplication of a graph matrix[8] over the embedding table as described in Section 3.2. It is worth noting that even in this setting, +0.7 average BLEU gains are obtained without introducing any extra trainable parameters. For simplicity, we name our graph-based approach *GraphMerge* and conduct experiments for 1, 2, and 3 hops.

It can be seen that GraphMerge consistently outperforms the baseline by a substantial margin: 1) the GraphMerge models yield better performance for all language directions, 2) GraphMerge with a 3-hop setting achieves the best results with an average gain of +2.3 BLEU. The largest gain of +3.5 BLEU can be found for into-English translations. Also, for out-of-English translation which is often considered the more difficult task, our method obtains an improvement of +1.0 BLEU.

A noteworthy finding is that as we increase the depth of the GNN from 1-hop to 3-hop, the performance consistently improves for almost all language pairs, resulting in notable gains of 1.6, 1.9, and 2.3 BLEU, respectively. We attribute the progressive improvement to the gradual enhancement of the quality of cross-lingual word embeddings. A more detailed analysis can be found in Section 5.1.

The results in other metrics, like sacreBLEU, ChrF++, and Comet can also be found in Table 11, where we show that our improvements remain consistent across a large spectrum of evaluation metrics.

---

[5]Patience is set to {20, 10}, i.e., training stops if performance on the validation set does not improve for the last {20, 10} checkpoints, with 1,000 steps between checkpoints.

[6]nrefs:1|case:mixed|eff:no|tok:13a|smooth:exp|version:2.3.1

[7]https://github.com/robertostling/eflomal

[8]In practice, for the graph $G$ in the naïve setting, we add an identity matrix $I$ to ensure no loss of information for current words.

| Model | High | | Medium | | Low | | ALL | | |
|---|---|---|---|---|---|---|---|---|---|
| | EN→X | X→EN | EN→X | X→EN | EN→X | X→EN | EN→X | X→EN | AVG |
| Baseline (Trans.-Big) | 28.7 | 31.3 | 31.0 | 31.4 | 20.0 | 25.6 | 26.5 | 29.4 | 28.0 |
| GraphMerge-1hop | 29.5 | 32.0 | 31.7 | 31.8 | 20.6 | 27.0 | 27.3 | 30.3 | 28.8 |
| GraphMerge-2hop | **29.7** | **32.2** | **32.0** | **32.0** | 20.9 | **27.4** | **27.6** | **30.5** | **29.1** |
| GraphMerge-3hop | 29.4 | 31.8 | 32.0 | 31.9 | **21.0** | 27.4 | 27.5 | 30.4 | 29.0 |
| 2-hop Gain | +1.0 | +0.9 | +1.0 | +0.6 | +0.9 | +1.8 | +1.1 | +1.1 | +1.1 |

Table 2: Large-scale experiments on the EC30 dataset (61M sentence pairs, 128K shared vocabulary). EN→X and X→EN denote the average performance of out-of- and into-English translation on each resource group, respectively. The best results in each column are in bold.

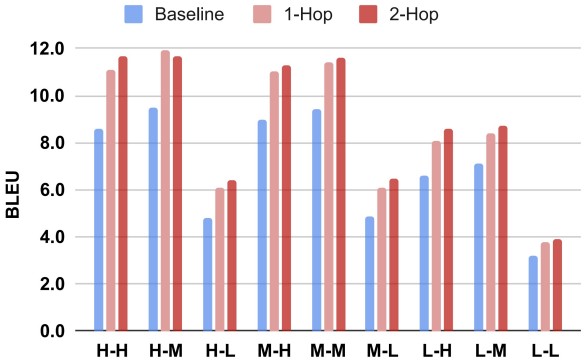

Figure 3: Zero-shot performance on EC30 (870 language directions), grouped by **H**igh-, **M**edium-, and **L**ow-resource.

| Settings | EN→X | X→EN | AVG |
|---|---|---|---|
| Baseline | 24.4 | 27.8 | 26.1 |
| 1-hop | 25.4 | 30.0 | 27.7 |
| 1-hop w/o Tie | 25.4 | 28.7 | 27.0 |
| 2-hop | 25.5 | 30.5 | 28.0 |
| 2-hop w/o Tie | 25.1 | 29.6 | 27.4 |
| 3-hop | 25.4 | 31.3 | 28.4 |
| 3-hop w/o Tie | 25.3 | 29.4 | 27.4 |
| 2-hop | 25.5 | 30.5 | 28.0 |
| *eflomal → FastAlign* | 25.4 | 30.1 | 27.8 |
| *intersect → gdfa* | 25.2 | 29.9 | 27.6 |

Table 3: Ablation experiment results on IWSLT14. 1-, 2-, and 3-hop refer to the number of hops in GraphMerge and 'w/o tie' indicates tied input and output with original embeddings instead of re-parameterized ones. *gdfa* is the abbreviation of *grow-diag-final-and*.

## 4.3 Large-scale Multilingual Results

To evaluate the scalability of our approach, we also run experiments on the EC30 dataset which includes 30 language pairs with 61M sentence pairs. Furthermore, we also use a bigger model (Transformer-Big) and larger vocabulary size (128K), resulting in a stronger baseline.

As shown in Table 2, our method achieves consistent improvements across all resource groups over the baseline for English-centric pairs. When extending GraphMerge from 1-hop to 2-hop, further improvements of up to +1.1 average BLEU points were achieved, but the improvements started to slightly weaken in the 3-hop setting. These improvements are in line with our results for IWSLT14, see Section 4.2. The results for individual languages can be found in Appendix A.4, where the results in other metrics, like SacreBLEU, ChrF++, and Comet are also reported and consistent improvements are shown as well.

We also evaluate the zero-shot translation performance on EC30 for 870 language directions not involving English. Figure 3 shows our results for 9 resource groups, considering all resource combinations (high, medium, low) for source and target. Note that here resource size refers to the amount of parallel English data that is available for a given source or target language. On average, our approach obtains improvements of 1.9 BLEU for all groups compared to the baseline.

## 4.4 Ablation

To investigate the impact of some specific settings in our framework, we conduct ablation experiments on the IWSLT14 dataset in this section.

**Tied Embeddings.** A basic setting for the transformer-based NMT model is to tie the decoder's input and output embedding (Press and Wolf, 2017; Pappas et al., 2018). Here, two embedding tables exist in our GraphMerge model, i.e., the original embedding table and the re-parameterized one. We test which one is better to tie with the decoder's output embedding. Note that all embeddings are trained from scratch.

**Graph Quality.** We use the *eflomal* alignment tool with the *intersect* strategy to extract alignments and construct the corresponding graph. The graphs induced from different alignment strategies may influence downstream results. We evaluate this point and set up experimental groups as follows: 1) Using *FastAlign* (Dyer et al., 2013) to extract alignments instead of using *eflomal*, where

| Model | EN↔DE | | EN↔NL | | EN↔AR | | EN↔HE | |
|---|---|---|---|---|---|---|---|---|
| | Similarity | BLEU | Similarity | BLEU | Similarity | BLEU | Similarity | BLEU |
| Baseline | 0.24 | 28.5 | 0.25 | 31.5 | 0.23 | 20.2 | 0.23 | 27.9 |
| GraphMerge-1hop | 0.35 | 30.2 | 0.37 | 33.4 | 0.32 | 21.7 | 0.32 | 30.0 |
| GraphMerge-2hop | 0.42 | 30.4 | 0.44 | 33.7 | 0.38 | 21.9 | 0.38 | 30.0 |
| GraphMerge-3hop | **0.46** | **30.7** | **0.48** | **34.0** | **0.41** | **22.4** | **0.41** | **30.1** |
| Model | EN↔ES | | EN↔FA | | EN↔PL | | EN↔IT | |
| | Similarity | BLEU | Similarity | BLEU | Similarity | BLEU | Similarity | BLEU |
| Baseline | 0.25 | 36.0 | 0.22 | 17.4 | 0.24 | 17.6 | 0.27 | 29.7 |
| GraphMerge-1hop | 0.38 | 37.5 | 0.31 | 19.0 | 0.35 | 18.8 | 0.40 | 31.3 |
| GraphMerge-2hop | 0.45 | 37.9 | 0.37 | 19.0 | 0.43 | 19.2 | 0.48 | 31.6 |
| GraphMerge-3hop | **0.49** | **38.2** | **0.40** | **19.9** | **0.47** | **19.4** | **0.52** | **32.2** |

Table 4: English-centric word similarity analysis for each language pair in the IWSLT14 dataset. A high degree of consistency between the similarity of representations and the corresponding BLEU scores can be found.

the latter is often considered to have better performance (Östling and Tiedemann, 2016). 2) Using *grow-diag-final-and* as the alignment strategy instead of *intersect*, which improves recall but reduces precision (Koehn, 2009).

Table 3 shows the ablation results on IWSLT14. We report results with different settings: 1-, 2-, 3-hop. By replacing the re-parameterized embeddings with the original one, BLEU performance drops by {0.7, 0.6, 1.0} on average, respectively. It shows that tying re-parameterized embeddings with the decoder's projection function significantly influences performance. The quality of the graph also mildly impacts downstream results: When alignments are noisier, as is the case for *FastAlign* and *gdfa*, performance drops by 0.2 and 0.4, respectively, compared to the 2-hop GraphMerge.

## 5  Analysis

### 5.1  English-Centric Cross-lingual Word Similarity

To verify whether embeddings of words with similar meanings are indeed closer to each other in the representation space, compared to the baseline, we conduct additional experiments as described below.

We utilize bilingual dictionaries from MUSE[9] (Lample et al., 2018a) as the ground truth and analyze the embedding similarity between word pairs. MUSE contains 110 English-centric bilingual dictionaries and all languages in our experiments are included. We limit our comparisons to the word pairs that exist in both our vocabulary of IWSLT14 and MUSE dictionaries. For each language pair, more than 1,000 word pairs exist in the intersection.

[9] https://github.com/facebookresearch/MUSE#ground-truth-bilingual-dictionaries

We use cosine similarity to evaluate the average distance between the word pairs across all language pairs. We also consider the isotropy of the space, which can be seen as the distribution bias of a space (Ethayarajh, 2019). Preserving isotropy is to avoid a situation where the similarities of words in a certain space are all significantly higher than those in other spaces, making the comparison across spaces not fair. The detailed results of the degree of isotropy can be found in Appendix A.6. In short, isotropies are all at a high level, i.e., low degree of anisotropy, and therefore not a cause for inflated similarities.

Table 4 shows the analysis results on IWSLT14. It is evident that a strong level of consistency exists in our findings: 1) Firstly, as we increase the depth of our GNN model from 1-hop to 3-hop, we observe a progressive increase in the similarities between word representations with similar meanings. This indicates that our re-parameterized method effectively enhances the cross-linguality of the embedding table. 2) Secondly, we observe consistent translation improvements in each direction as the cross-linguality becomes stronger. This supports our hypothesis that improving the multilinguality of the shared embedding table results in greater translation quality.

### 5.2  Beyond English-Centric Cross-lingual Word Similarity

As mentioned above, for practicality, we choose English as the pivot and only mine equivalent words between English and other languages. Hence, in non-English directions, such as DE-NL, meaning-equivalent relationships are not explicitly incorporated into the graph (e.g., the empty parts in Figure 2). We argue that even in this setting, cross-

| Model | DE↔NL | DE↔AR | DE↔HE | NL↔AR | NL↔HE | AR↔HE |
|---|---|---|---|---|---|---|
| Baseline | 0.29 | 0.23 | 0.25 | 0.24 | 0.26 | 0.29 |
| GraphMerge-1hop | 0.36 | 0.28 | 0.30 | 0.30 | 0.31 | 0.33 |
| GraphMerge-2hop | 0.42 | 0.32 | 0.34 | 0.35 | 0.35 | 0.37 |
| GraphMerge-3hop | **0.47** | **0.36** | **0.38** | **0.39** | **0.39** | **0.41** |

Table 5: Beyond English-centric word similarity analysis for each language pair in the IWSLT14 dataset. Consistently enhanced multilinguality shows as graph networks go deeper.

| Model | EN→DA | DA→EN | EN→AF | AF→EN |
|---|---|---|---|---|
| Bilingual Baseline | 35.3 | 35.4 | 31.7 | 35.9 |
| Bilingual GraphMerge-3hop | **35.6** | **35.8** | **33.9** | **39.6** |
| EC30 Baseline | 36.9 | 39.7 | 38.4 | 48.2 |
| EC30 GraphMerge-3hop | **38.1** | **40.0** | **38.5** | **50.1** |

Table 6: Results of bilingual experiments on EN-DE and EN-AF language pairs. The evaluation set is the same as that of EC30, i.e., Flores-101. The best results in each column are in bold.

linguality could still be enhanced due to the pivot of English bridge the way of knowledge passing from German to Dutch through our multi-hop mechanism. We evaluate this point as follows.

Non-English-centric dictionaries included in MUSE are limited, therefore we extend MUSE by mapping words paired with the same English words together, e.g., given an EN-DE pair {*bike*, *Fahrrad*} and an EN-NL pair {*bike*, *fiets*}, we can build a new word pair {*Fahrrad*, *fiets*} for DE-NL. For the 8 languages among the IWSLT14 dataset, we build 28 dictionaries for the corresponding non-English-centric language pairs. Each of these dictionaries contains more than 1,000 non-English-centric word pairs except for AR-HE, where 697 pairs are found.

We report analysis results for 6 language pairs among 4 languages, involving the same and different writing systems, i.e., DE, NL, AR, and HE in Table 5. One can easily see that our reparameterized embeddings consistently exhibit a higher degree of cross-linguality compared to the baseline, which underlines the generality of our approach even if only English-centric equivalence relationships are leveraged. Full results for 8 languages and 28 pairs can be found in Appendix A.7.

### 5.3 Bilingual Experiments

In this section, we explore whether our method is able to bring benefits for the "extreme" setting of multilingual translation systems, i.e., bilingual translation. We argue that the advantages of better semantic alignments should also apply here.

We pick two language pairs from EC30, i.e., EN-DA (1M) and EN-AF (100K), and conduct experiments individually (note that the graph is rebuilt for each as well). We use a rule of thumb setting, i.e.,

transformer-base (6 layers, 512 embeddings dim, 1024 feedforward, 4 attention heads), as the backbone. Also, we shrunk the vocabulary size to 16K considering less data. In Table 6, we compare the bilingual baseline with the GraphMerge-3hop setting in 4 language directions, where we also list the performances in these directions from our MMT model trained on EC30 (see Section 4.3).

It shows that even in an "extreme" setting (bilingual translation), our method still brings clear benefits, especially for the low-resource pair (EN-AF), +2.2 and +3.7 BLEU gains are achieved, respectively. When we extend the "extreme" setting to 30-language MMT, resulting in a stronger baseline, the improvements remain in these four directions.

It is worth noting that in the bilingual setting, only one language is fed to the encoder or decoder. We attribute the improvements here to two potential factors: 1) Monolingual synonyms are modeled better. E.g., *bike* and *bicycle* are both linked to the pivot words (e.g., *cykel* in Danish), resulting in enhanced representational similarity via the multi-hop mechanism, and 2) the enhanced cross-lingual word similarity, e.g., that of *bike* and *cykel*, may also lead to higher accuracy during decoding. We leave the further analysis for future work.

In addition, we also demonstrate our method's applicability for bilingual translation on the English-Hebrew track in the WMT 2023 competition. In short, clear gains are still observed, even when tested on a large-scale dataset exceeding 30 million sentences. A brief summary of partial experiments results for our WMT2023 submission (Wu et al., 2023) can be found in Section A.5.

| Model | WPS | Times |
|---|---|---|
| Transformer (30K) | 201,378 | 1.00 |
| GraphMerge-1hop | 192,367 | 1.04 |
| GraphMerge-2hop | 188,851 | 1.06 |
| Transformer-Big (128K) | 69,702 | 1.00 |
| GraphMerge-1hop | 43,416 | 1.61 |
| GraphMerge-2hop | 33,912 | 2.05 |

| Model | Params | Times |
|---|---|---|
| Transformer (30K) | 62.3M | 1.00 |
| GraphMerge-1hop | 63.3M | 1.01 |
| GraphMerge-2hop | 64.4M | 1.02 |
| Transformer-Big (128K) | 438.6M | 1.00 |
| GraphMerge-1hop | 442.8M | 1.01 |
| GraphMerge-2hop | 447.0M | 1.02 |

Table 7: "WPS" is the average word number the model processes per second. We fix "Time" of transformer models to 1.0. 30K and 128K indicate the corresponding vocabulary size.

## 5.4 Speed and Memory

Compared with the standard transformer model, our method introduces extra graph operations. Particularly, a big graph (adjacency) matrix of size $30K \times 30K$ is multiplied with the original embedding table. An obvious question here is whether training latency increases to an unacceptable level. We show training latency in Table 7. All experiments here were conducted on a single NVIDIA A6000 GPU card with FP16 optimization. We utilized the sparse matrix optimization provided by PyTorch (Paszke et al., 2019) for implementation.

As shown in Table 7, for the model we used for IWSLT14 (30K vocabulary), the extra training latency is negligible, with only 4% and 6% for the 1-hop and 2-hop models. Even for a much bigger graph matrix, (128K vocabulary, nearly 16 times bigger adjacency matrix), training latency is limited. The only trainable parameter we introduced here is the dense matrix within the graph networks, which constitutes approximately 2% of all model parameters. Furthermore, due to the sparsity of the adjacency matrix, the additional memory usage is negligible, amounting to less than 1%.

It is worth noting that although the graph matrix can be sizable, the re-parameterized embedding table can be decoupled and stored for online deployment, meaning inference latency is exactly the same as for the baseline. Meanwhile, for training, GraphMerge only applies to the embedding table once per training step. In other words, the batch size or the model size of the transformer architecture does not matter. Both benefits show

practicality for large-scale settings.

## 6 Conclusions

In this paper, we target the shortcomings of solely relying on a shared vocabulary for knowledge transfer. Broadly speaking, our approach is to mine word equivalence across languages and inject such priors into the embedding table using a graph network, thereby significantly improving transfer.

Our experiments show that our approach results in embeddings with a higher degree of multilinguality, leading to consistent improvements in MNMT. E.g., our approach achieves 2.3 average BLEU gain on IWSLT14, and the improvements still hold even on a much larger and more diverse dataset, namely EC30 (61M bitext, 30 language pairs). Also, even for the minimal setting of MMT, i.e., bilingual translation, the performance gains are still held for 30+M dataset.

At the same time, our framework remains practical: 1) Through the multi-hop mechanism, the pivot language (English) bridges the way of knowledge transfer among non-English language pairs. Therefore, even when only English-centric bitext datasets are available, multilingual transfer can be achieved. 2) A negligible number of additional trainable parameters are required with a limited increase in computational costs during training. Meanwhile, the inference latency is exactly as same as benchmark systems by storing and deploying the re-parameterized embeddings for online systems.

## 7 Limitations

As the MT system scales up, a big vocabulary may be introduced. Extra computing costs may get pronounced as the vocabulary scales to a super big size, e.g., NLLB (Costa-jussà et al., 2022) uses a 256K vocabulary.

## 8 Broader Impact

Multilingual translation systems have significant progress recently. However, potential challenges such as mistranslation or off-target issues still exist. Moreover, the fairness problem also raise, i.e., the generation ability is not guaranteed to be fair across languages or demographic features, which may run the risk of exploiting and reinforcing the societal biases (e.g. gender or race bias) that are present in the underlying data.

## 9 Acknowledgement

We thank the anonymous reviewers for their efforts to make this paper better. We thank our colleagues from the Language Technology Lab at the University of Amsterdam, especially Shaomu Tan, Vlad Niculae, and David Stap, for their suggestions on data, code, and writing. We thank Chongyang for its invaluable spiritual support. This research was funded in part by the Netherlands Organization for Scientific Research (NWO) under project number VI.C.192.080.

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

## A Appendix

### A.1 Dataset Details

The detailed data size of IWSLT14 are as follows: FA (89K), PL (128K), AR (140K), HE (144K), NL (153K), DE (160K), IT (167K), and ES (169K)

We list the details of the EC30 dataset in Table 8. Overall, EC30 is an English-centric multilingual machine translation dataset containing 61 million sentences including 31 languages (together with

| High | de | nl | fr | es | ru | cs | hi | bn | ar | he |
|------|----|----|----|----|----|----|----|----|----|----|
| Medium | sv | da | it | pt | pl | bg | kn | mr | mt | ha |
| Low | af | lb | ro | oc | uk | sr | sd | gu | ti | am |

Table 8: Languages grouped in different resource levels in EC30. We use ISO 639-1 in this table.

| Model | Sampled Data (offline) | | Full Data (offline) | | Full Data (online) | |
|-------|:-----:|:-----:|:-----:|:-----:|:-----:|:-----:|
| | EN→HE | HE→EN | EN→HE | HE→EN | EN→HE | HE→EN |
| Bilingual Baseline | 24.6 | 31.1 | 34.1 | 46.0 | - | - |
| MMT Baseline | 24.7 | 31.6 | 34.1 | 45.8 | 33.3 | 50.3 |
| MMT + GraphMerge 1-hop | **26.2** | **32.3** | **34.3** | **46.2** | **33.6** | **50.7** |

Table 9: Offline and online evaluation results on sampled (2M) and full (34M) training data for the WMT2023 competition. MMT + GraphMerge 1-hop means that we equip graph-based re-parameterized embedding tables for our MMT baseline. The best BLEU scores in each column are written in bold.

English). EC30 is more profound in the total number of languages and in the balance of language family and writing systems. Specifically, for each language family, we include 6 representative languages across different resources (2 for each resource level). we categorized the languages into high-, medium-, and low-resource groups, each having 5 million, 1 million, and 100,000 bitext examples, respectively.

For experiments on EC30, we use temperature sampling to select data for training BPE due to it being highly imbalanced (high resource data is 50 times bigger than low ones), the temperature is aligned with that of the MNMT training phase. More specifically, we keep the lowest-resource dataset unchanged and sample data in other bitexts to match the temperature ratio. Then, we merge the data for BPE training and apply BPE to the EC30 corpus.

## A.2 Training Settings

For all of our experiments, the training parameters of our methods and baselines range from 62M to 447M. All models are trained on 4 A6000 GPU cards and training times are from 10 hours to 8 days.

## A.3 Detailed IWSLT14 Results

We show full results on IWSLT14 in tokenized BLEU in Table 10. An interesting finding is that as graph networks go deeper (from 1-hop to 5-hop), into-English translation consistently gets improved, while the average best one for out-of-English translation is GraphMeger with a 3-hop setting.

We also show full results on IWSLT14 in other wide-used metrics, like SacreBLEU, ChrF++, and Comet in Table 11. This shows that our improve-

ments remain consistent across a large spectrum of evaluation metrics.

## A.4 Detailed EC30 Results

We show full English-centric results (60 directions) on the EC30 dataset in Table 12. The results are placed from high-, medium-, to low-resource groups.

Also, we show the results in SacreBLEU, CharF++, and Comet on EC30 in Table 13. It shows that our improvements remain consistent across a large spectrum of evaluation metrics.

## A.5 Detailed WMT2023 submission

For our WMT 2023 submission (Wu et al., 2023), we make use of all the available data from the constrained track of the shared task for English-Hebrew translation, where 70M parallel data are released. After data preprocessing, we reduced the size of the bitext to 34M. We further normalize and tokenize bitexts using Moses and SentencePiece with 32K vocabulary size.

For offline evaluation, we conducted experiments on full data (34M) and the sampled data (2M). We apply Transformer-Large[10] as the backbone for the full dataset considering the large scale of bilingual sentences while using Transformer Base for the sampled dataset here. Both bilingual and MMT (two directions within one model) settings are tested. We further equip GraphMerge-1hop upon MMT baselines to show the benefits, and the results can be found in Table 9.

It is easy to see that 1) The performances of the bilingual and MMT models are comparable on both full and sampled datasets. 2) Moreover, the

---

[10]12-layer encoder and decoder, 16 attention heads, 1024 embedding dimensions, and 4096 feed-forward dimensions.

| Model | EN→DE | EN→ES | EN→FA | EN→AR | EN→HE | EN→NL | EN→PL | EN→IT |
|---|---|---|---|---|---|---|---|---|
| Our baseline | 27.9 | 37.2 | 14.7 | 14.1 | 24.6 | 31.8 | 15.3 | 29.7 |
| GraphMerge-1hop | **29.2** | **38.2** | 15.9 | 14.6 | **25.6** | 33.0 | 16.1 | 31.0 |
| GraphMerge-2hop | 29.1 | **38.2** | 16.1 | **14.7** | 25.4 | **32.8** | **16.2** | 31.2 |
| GraphMerge-3hop | 28.9 | 37.9 | **16.2** | **14.7** | 25.2 | 32.6 | **16.2** | **31.3** |
| GraphMerge-4hop | 28.9 | 37.9 | 16.1 | 14.5 | 25.0 | **32.8** | 16.0 | 31.2 |
| GraphMerge-5hop | 28.3 | 37.4 | 15.6 | 14.0 | 24.6 | 32.3 | 15.6 | 30.5 |

| Model | DE→EN | ES→EN | FA→EN | AR→EN | HE→EN | NL→EN | PL→EN | IT→EN |
|---|---|---|---|---|---|---|---|---|
| Our baseline | 29.2 | 34.9 | 20.1 | 26.4 | 31.2 | 31.2 | 19.8 | 29.6 |
| GraphMerge-1hop | 31.2 | 36.9 | 22.1 | 28.8 | 34.4 | 33.8 | 21.6 | 31.6 |
| GraphMerge-2hop | 31.8 | 37.6 | 21.9 | 29.2 | 34.7 | 34.6 | 22.1 | 32.1 |
| GraphMerge-3hop | 32.4 | 38.5 | 23.6 | 30.0 | 35.1 | 35.4 | 22.6 | 33.0 |
| GraphMerge-4hop | 32.8 | 38.9 | 24.1 | **30.3** | 35.9 | 35.8 | **22.7** | 33.1 |
| GraphMerge-5hop | **33.5** | **39.4** | **24.3** | **30.3** | **36.0** | **36.2** | **22.7** | **33.5** |

Table 10: Detailed results on the IWSLT14 dataset. We report the BLEU score for each English-centric translation direction.

| SacreBLEU | DE | ES | FA | AR | HE | NL | PL | IT | EN→X | X→EN | AVG |
|---|---|---|---|---|---|---|---|---|---|---|---|
| baseline | 27.6 | 35.1 | 16.8 | 19.5 | 27.2 | 30.8 | 17.1 | 29.0 | 24.7 | 26.0 | 25.4 |
| GraphMerge-1hop | 29.7 | 36.9 | 18.7 | 21.3 | 29.4 | 32.9 | 18.5 | 30.6 | **25.5** | 29.0 | 27.2 |
| GraphMerge-2hop | 29.7 | 37.1 | 18.7 | 21.1 | 29.1 | 33.0 | 18.5 | 30.8 | 25.2 | 29.3 | 27.3 |
| GraphMerge-3hop | **30.1** | **37.6** | **19.5** | **21.9** | **29.5** | **33.5** | **19.0** | **31.5** | 25.3 | **30.3** | **27.8** |
| 3-hop Gain | +2.6 | +2.6 | +2.7 | +2.4 | +2.3 | +2.8 | +1.9 | +2.5 | +0.6 | +4.3 | +2.4 |
| **ChrF++** | DE | ES | FA | AR | HE | NL | PL | IT | EN→X | X→EN | AVG |
| baseline | 52.1 | 58.0 | 40.1 | 43.7 | 50.5 | 54.5 | 42.1 | 52.8 | 50.0 | 48.4 | 49.2 |
| GraphMerge-1hop | 54.0 | 60.0 | 41.8 | 45.4 | 52.5 | 56.5 | 43.6 | 54.6 | **50.6** | 51.5 | 51.0 |
| GraphMerge-2hop | 54.0 | 60.0 | 41.8 | 45.4 | 52.5 | 56.5 | 43.6 | 54.6 | 50.6 | 51.5 | 51.0 |
| GraphMerge-3hop | **54.3** | **60.3** | **42.2** | **45.9** | **52.9** | **56.8** | **43.9** | **55.0** | 50.4 | **52.4** | **51.4** |
| 3-hop Gain | +2.3 | +2.3 | +2.1 | +2.2 | +2.4 | +2.3 | +1.8 | +2.2 | +0.4 | +4.0 | +2.2 |
| **Comet** | DE | ES | FA | AR | HE | NL | PL | IT | EN→X | X→EN | AVG |
| baseline | 75.5 | 79.8 | 74.2 | 76.2 | 78.8 | 78.7 | 72.9 | 78.3 | 78.9 | 74.7 | 76.8 |
| GraphMerge-1hop | 77.5 | 81.4 | 76.2 | 78.2 | 80.7 | 80.8 | 74.8 | 80.0 | **79.9** | 77.4 | 78.7 |
| GraphMerge-2hop | 77.5 | 81.7 | 76.1 | 78.3 | 80.9 | 80.9 | 74.8 | 80.2 | 79.8 | 77.8 | 78.8 |
| GraphMerge-3hop | **77.8** | **81.8** | **76.6** | **78.5** | **81.2** | **81.3** | **75.2** | **80.5** | 79.7 | **78.5** | **79.1** |
| 3-hop Gain | +2.3 | +2.1 | +2.4 | +2.3 | +2.4 | +2.6 | +2.3 | +2.2 | +0.8 | +3.8 | +2.3 |

Table 11: Detailed results on the IWSLT14 dataset in SacreBLEU, ChrF++, and Comet. Consistent Gains can be found for the GraphMerge-3hop setting. Especially, into-English improvements are large.

models equipped with GraphMerge-1hop achieve consistent improvements in both two directions, especially for sampled data, the gain is evident.

Finally, we use the model trained on full data for WMT23's final online evaluation. It shows that even on a 30+M dataset consisting of two languages only, GraphMerge-1hop can still achieve +0.3 and +0.4 BLEU improvements for out-of- and into-English translation, which is also consistent with offline evaluation. Other strategies we used in this competition can be found in (Wu et al., 2023).

## A.6 Isotropy Definition and Results

Introducing isotropy is to avoid a situation where the similarities of words in a certain space are all significantly higher than those in other spaces making it difficult to compare results across spaces.

Given the degree of isotropy derived from the standard transformer model as an example:

For a set of word pairs (such as EN-DE), given the degree of isotropy derived from a specific model as an example: 1) for each word $x_i$ in the intersection of MUSE EN-DE dictionary and our vocabulary, we randomly select the target word $x_j$ 50 times from the whole subword space and average the similarity score between the embedding of $x_i$ and $x_j$ as $I_i$, 2) then, average $I_i$ for all the words $x_i$ in the subsection as the metric $I_{EN-DE}$. For the optimal isotropic space, such a metric should be close to zero. Meanwhile, if a large difference in isotropy between two spaces exists, the similarity comparison across space is problematic.

Table 14 shows the analysis results on IWSLT14. For all of the 8 language pairs, more than 1,000

| Model | EN→DE | EN→NL | EN→FR | EN→ES | EN→RU | EN→CS | EN→HI | EN→BN | EN→AR | EN→HE |
|---|---|---|---|---|---|---|---|---|---|---|
| baseline | 32.1 | 24.1 | 41.8 | 22.8 | 22.7 | 25.9 | 39.9 | 32.2 | 19.5 | 25.8 |
| GraphMerge-1hop | 32.8 | **24.7** | 43.1 | 23.6 | 24.0 | 26.1 | 41.2 | **32.9** | 19.7 | 26.8 |
| GraphMerge-2hop | **33.3** | 24.6 | **43.3** | **23.8** | **24.3** | **26.6** | **41.5** | 32.3 | **20.3** | **26.9** |
| GraphMerge-3hop | 33.2 | 24.6 | 42.9 | 23.6 | 23.8 | 26.2 | 41.0 | 32.7 | 19.6 | 26.8 |

| Model | DE→EN | NL→EN | FR→EN | ES→EN | RU→EN | CS→EN | HI→EN | BN→EN | AR→EN | HE→EN |
|---|---|---|---|---|---|---|---|---|---|---|
| baseline | 37.1 | 27.9 | 37.1 | 24.9 | 27.9 | 32.7 | 33.7 | 28.1 | 27.9 | 35.5 |
| GraphMerge-1hop | **38.1** | 28.4 | 37.6 | 25.5 | 28.8 | **33.6** | 34.4 | 28.9 | 28.4 | 36.5 |
| GraphMerge-2hop | 38.0 | **28.9** | **38.0** | **25.8** | **28.9** | 33.3 | 34.3 | **29.5** | **28.6** | **36.6** |
| GraphMerge-3hop | 37.8 | 28.2 | 37.4 | 25.5 | 28.6 | 33.6 | 33.5 | 28.5 | 28.6 | 36.5 |

| Model | EN→SV | EN→DA | EN→IT | EN→PT | EN→PL | EN→BG | EN→KN | EN→MR | EN→MT | EN→HA |
|---|---|---|---|---|---|---|---|---|---|---|
| baseline | 36.0 | 36.9 | 24.6 | 34.6 | 16.7 | 36.1 | 34.9 | 27.3 | 48.4 | 14.4 |
| GraphMerge-1hop | 36.6 | 37.5 | 25.4 | **35.3** | 17.3 | 37.8 | 35.3 | 27.7 | 49.7 | **14.5** |
| GraphMerge-2hop | 36.9 | **38.3** | 25.9 | 35.2 | **17.9** | 37.9 | 35.8 | 27.9 | **50.1** | 14.5 |
| GraphMerge-3hop | **37.1** | 38.1 | 25.3 | 35.3 | 17.8 | 37.6 | **36.2** | **28.0** | 49.9 | 14.4 |

| Model | SV→EN | DA→EN | IT→EN | PT→EN | PL→EN | BG→EN | KN→EN | MR→EN | MT→EN | HA→EN |
|---|---|---|---|---|---|---|---|---|---|---|
| baseline | 39.5 | 39.7 | 27.3 | 39.9 | 23.3 | 35.0 | 24.6 | 26.1 | 48.2 | 10.2 |
| GraphMerge-1hop | **40.3** | 40.2 | 27.8 | **40.4** | 23.7 | 35.2 | 24.8 | 26.5 | 49.0 | **10.5** |
| GraphMerge-2hop | 40.2 | **40.5** | **28.0** | 40.2 | **24.0** | **36.1** | 25.3 | 27.1 | **49.5** | 9.3 |
| GraphMerge-3hop | 39.9 | 40.0 | 27.9 | 40.2 | 23.9 | 35.6 | 25.0 | 26.9 | 49.5 | 9.8 |

| Model | EN→AF | EN→LB | EN→RO | EN→OC | EN→UK | EN→SR | EN→SD | EN→GU | EN→TI | EN→AM |
|---|---|---|---|---|---|---|---|---|---|---|
| baseline | 38.4 | 16.2 | 25.9 | 29.3 | 21.1 | 24.1 | 5.0 | 28.5 | 4.4 | 7.0 |
| GraphMerge-1hop | 38.3 | 16.1 | 26.2 | 30.5 | 22.8 | 25.1 | 5.5 | 29.6 | 4.5 | 7.8 |
| GraphMerge-2hop | **38.8** | **17.0** | 26.7 | 29.8 | 23.1 | **25.6** | **5.9** | 29.8 | **4.7** | **8.0** |
| GraphMerge-3hop | 38.5 | 16.5 | **27.7** | **30.7** | **23.4** | 25.5 | 5.1 | **29.9** | 4.7 | 7.8 |

| Model | AF→EN | LB→EN | RO→EN | OC→EN | UK→EN | SR→EN | SD→EN | GU→EN | TI→EN | AM→EN |
|---|---|---|---|---|---|---|---|---|---|---|
| baseline | 48.2 | 21.8 | 31.0 | 35.7 | 27.3 | 30.6 | 1.4 | 24.4 | 15.1 | 20.6 |
| GraphMerge-1hop | 49.7 | **23.2** | 32.0 | 38.1 | 28.4 | 32.1 | **2.5** | 26.1 | 15.7 | 22.4 |
| GraphMerge-2hop | **50.2** | 22.4 | **32.5** | **39.1** | **29.2** | **32.8** | 1.2 | 27.0 | **16.4** | 22.8 |
| GraphMerge-3hop | 50.1 | 23.1 | 32.1 | 38.7 | 29.1 | 32.7 | 1.5 | **27.4** | 16.1 | **22.9** |

Table 12: Detailed results on the EC30 dataset in 60 directions. The results are placed from high-, medium-, to low-resource groups. We report the BLEU score for each direction.

word pairs exist in the intersection of our vocabulary and MUSE dictionaries. It is easy to see that our re-parameterized embeddings are consistent with better cross-linguality, i.e., the distance between word representations with similar meanings is much smaller. Meanwhile, the isotropies are all at a high level (low degree), i.e., evaluation risk due to space bias is excluded as well. In other words, similarity comparisons in Table 4 are fair.

## A.7 Full Results of Beyond English-Centric Word Similarity

In Table 15, we show the full results of non-English-centric word similarity analysis for 28 language directions in the IWSLT14 dataset. It is easy to find that, as graph networks go deeper, word similarity consistently gets closer.

| SacreBLEU | High | | Medium | | Low | | ALL | | |
|---|---|---|---|---|---|---|---|---|---|
| | EN→X | X→EN | EN→X | X→EN | EN→X | X→EN | EN→X | X→EN | AVG |
| Baseline (Trans.-Big) | 28.4 | 31.1 | 29.8 | 31.2 | 19.4 | 25.5 | 25.9 | 29.3 | 27.6 |
| GraphMerge-1hop | 29.1 | 31.6 | 30.4 | 31.5 | 20.0 | 26.7 | 26.5 | 29.9 | 28.2 |
| GraphMerge-2hop | **29.3** | **31.9** | **30.8** | **31.7** | 20.3 | 27.1 | **26.8** | **30.2** | **28.5** |
| GraphMerge-3hop | 29.2 | 31.6 | 30.7 | 31.7 | **20.4** | **27.3** | 26.8 | 30.2 | 28.5 |
| 2-hop Gain | +0.9 | +0.8 | +1.0 | +0.5 | +0.8 | +1.6 | +0.9 | +1.0 | +0.9 |

| ChrF++ | High | | Medium | | Low | | ALL | | |
|---|---|---|---|---|---|---|---|---|---|
| | EN→X | X→EN | EN→X | X→EN | EN→X | X→EN | EN→X | X→EN | AVG |
| Baseline (Trans.-Big) | 52.5 | 57.1 | 53.9 | 56.4 | 42.8 | 49.7 | 49.8 | 54.4 | 52.1 |
| GraphMerge-1hop | 53.0 | 57.5 | 54.4 | 56.6 | 43.5 | 50.6 | 50.3 | 54.9 | 52.6 |
| GraphMerge-2hop | **53.3** | **57.6** | **54.8** | **56.6** | **43.8** | 50.9 | **50.6** | **55.0** | **52.8** |
| GraphMerge-3hop | 53.2 | 57.5 | 54.7 | 56.6 | 43.8 | **51.0** | 50.6 | 55.0 | 52.8 |
| 2-hop Gain | +0.7 | +0.5 | +0.9 | +0.2 | +1.0 | +1.2 | +0.9 | +0.6 | +0.8 |

| Comet | High | | Medium | | Low | | ALL | | |
|---|---|---|---|---|---|---|---|---|---|
| | EN→X | X→EN | EN→X | X→EN | EN→X | X→EN | EN→X | X→EN | AVG |
| Baseline (Trans.-Big) | 82.0 | 83.4 | 80.6 | 79.7 | 73.3 | 72.7 | 78.6 | 78.6 | 78.6 |
| GraphMerge-1hop | 82.8 | 83.9 | 81.2 | 80.2 | 74.2 | 74.0 | 79.4 | 79.4 | 79.4 |
| GraphMerge-2hop | **83.1** | **84.0** | **81.5** | **80.2** | 74.6 | 74.3 | **79.8** | **79.5** | **79.6** |
| GraphMerge-3hop | 83.0 | 83.9 | 81.5 | 80.1 | **74.8** | **74.5** | 79.7 | 79.5 | 79.6 |
| 2-hop Gain | +1.1 | +0.7 | +0.9 | +0.5 | +1.4 | +1.6 | +1.1 | +0.9 | +1.0 |

Table 13: Large-scale experiments on the EC30 dataset (61M sentence pairs, 128K shared vocabulary) in Sacre-BLEU, ChrF++, and Comet. EN→X and X→EN denote the average performance of out-of- and into-English translation on each resource group, respectively.

| Model | EN-DE | EN-ES | EN-FA | EN-AR | EN-HE | EN-NL | EN-PL | EN-IT |
|---|---|---|---|---|---|---|---|---|
| Baseline | 0.068 | 0.071 | 0.074 | 0.073 | 0.072 | 0.069 | 0.074 | 0.073 |
| GraphMerge-1hop | 0.001 | -0.001 | -0.001 | -0.001 | -0.001 | -0.001 | -0.001 | -0.002 |
| GraphMerge-2hop | 0.001 | 0.001 | 0.002 | 0.001 | 0.001 | 0.001 | 0.002 | 0.001 |
| GraphMerge-3hop | 0.001 | 0.001 | 0.001 | 0.001 | 0.001 | 0.001 | 0.001 | 0.001 |

Table 14: The degree of isotropy in each space. It is easy to find that all of the degrees induces from our spaces are close to 0, i.e., evaluation risk due to space bias is excluded.

| Model | DE↔NL | DE↔AR | DE↔HE | DE↔ES | DE↔FA | DE↔PL | DE↔IT |
|---|---|---|---|---|---|---|---|
| baseline | 0.30 | 0.23 | 0.25 | 0.27 | 0.25 | 0.27 | 0.29 |
| GraphMerge-1hop | 0.37 | 0.28 | 0.30 | 0.34 | 0.30 | 0.33 | 0.36 |
| GraphMerge-2hop | 0.43 | 0.33 | 0.34 | 0.40 | 0.35 | 0.39 | 0.42 |
| GraphMerge-3hop | **0.47** | **0.36** | **0.38** | **0.44** | **0.38** | **0.43** | **0.46** |
| Model | NL↔AR | NL↔HE | NL↔ES | NL↔FA | NL↔PL | NL↔IT | AR↔HE |
| baseline | 0.24 | 0.26 | 0.28 | 0.25 | 0.28 | 0.30 | 0.29 |
| GraphMerge-1hop | 0.30 | 0.31 | 0.36 | 0.30 | 0.35 | 0.39 | 0.33 |
| GraphMerge-2hop | 0.35 | 0.35 | 0.42 | 0.35 | 0.41 | 0.45 | 0.38 |
| GraphMerge-3hop | **0.39** | **0.39** | **0.46** | **0.39** | **0.45** | **0.49** | **0.41** |
| Model | AR↔ES | AR↔FA | AR↔PL | AR↔IT | HE↔ES | HE↔FA | HE↔PL |
| baseline | 0.26 | 0.27 | 0.25 | 0.27 | 0.26 | 0.26 | 0.27 |
| GraphMerge-1hop | 0.32 | 0.32 | 0.31 | 0.33 | 0.32 | 0.31 | 0.31 |
| GraphMerge-2hop | 0.37 | 0.36 | 0.36 | 0.39 | 0.36 | 0.35 | 0.36 |
| GraphMerge-3hop | **0.40** | **0.40** | **0.39** | **0.43** | **0.40** | **0.39** | **0.40** |
| Model | HE↔IT | ES↔PA | ES↔PL | ES↔IT | FA↔PL | FA↔IT | PL↔IT |
| baseline | 0.27 | 0.25 | 0.28 | 0.33 | 0.24 | 0.25 | 0.29 |
| GraphMerge-1hop | 0.33 | 0.31 | 0.36 | 0.41 | 0.29 | 0.31 | 0.37 |
| GraphMerge-2hop | 0.38 | 0.36 | 0.41 | 0.48 | 0.34 | 0.36 | 0.42 |
| GraphMerge-3hop | **0.42** | **0.40** | **0.45** | **0.52** | **0.37** | **0.39** | **0.47** |

Table 15: Full results of non-English-centric word similarity analysis on the IWSLT14 dataset for 28 language pairs.