# OpenReview forum: "Beyond Shared Vocabulary: Increasing Representational Word Similarities across Languages for Multilingual Machine Translation"
_EMNLP/2023/Conference — EMNLP 2023 Main_

### Official Review · Reviewer_hsEL · 2023-08-04

**Soundness:** 4

**Excitement:**

3: Ambivalent: It has merits (e.g., it reports state-of-the-art results, the idea is nice), but there are key weaknesses (e.g., it describes incremental work), and it can significantly benefit from another round of revision. However, I won't object to accepting it if my co-reviewers champion it.

**Paper Topic And Main Contributions:**

This paper improves multilingual MT by (1) constructing a word similarity graph from word-aligned parallel text and (2) inserting a graph neural network after the word embeddings of the MT system. Improvements are large (2+ BLEU).

**Questions For The Authors:**

A. How do these BLEU scores compare with bilingual MT system scores?

B. You use MUSE for evaluating your word embeddings, but couldn't you construct a word similarity graph directly from MUSE?

C. Have you considered building graphs for other relationships besides synonymy? For example, if in French word A is a hyponym of word B, and in English word C is a hyponym of word D, and we know that word A and word C are translations of each other, could the GNN somehow infer that word B and word D are more likely to be translations of each other?


**Reasons To Accept:**

The model is based on an intuitive idea, that word similarity should be transitive, and the improvements are large (2+ BLEU).

**Reasons To Reject:**

I have some hesitation about including a word-similarity graph in an MT system, for two reasons.

The first is that the idea that the semantic similarity of words across languages can be quantified is complicated by a few facts:
- Because of polysemy, some words have multiple meanings
- Because of subword segmentation (line 159), some "words" do not have meanings at all

The second may be merely aesthetic; one of the benefits of neural MT over phrase-based MT was that it was a single end-to-end model, without separate stages for things like word alignment. This work brings back a separate word-alignment stage.

Not reasons to reject, but weaknesses:
- Please state earlier (line 187) what word alignment models are used, since this is part of your model. Also, please state explicitly what training data is used for the word alignment models.
- BLEU should be computed on untokenized references (e.g., using SacreBLEU), not tokenized references.
- Another metric like BLEU would be helpful.

**Reproducibility:**

4: Could mostly reproduce the results, but there may be some variation because of sample variance or minor variations in their interpretation of the protocol or method.

**Reviewer Confidence:**

4: Quite sure. I tried to check the important points carefully. It's unlikely, though conceivable, that I missed something that should affect my ratings.

---

> ### Author Rebuttal · Authors · 2023-08-28
>
> Thanks for your efforts on our paper. We summarize and answer your questions as follows:
>
> >[Reason to Reject-1]: Semantic similarity of words across languages can be quantified is complicated by a few facts: 1) some words have multiple meanings, and 2) some "subwords" may not have meanings at all.
>
> **[Response-1]:** Yes, this is a good point and we also considered these during this project:
> 1) Firstly, polysemy is not the only challenge for our method. Actually, it applies to the fact that we only use one embedding to represent the semantics of a word. The standard design of shared vocabulary also has this problem (Lines 65-67) and here we do not focus on this in this paper (lines 68-69).
> 2) Secondly, we agree that some “short subwords” may not have explicit meaning. However, the information transfer between two “short subwords” may also help with modeling morphological aspects across languages.
>
> >[Reason to Reject-2]: A separate word-alignment stage is not aesthetic enough.
>
> **[Response-2]:** We accept this point.
>
> >[Weaknesses-3]: BLEU should be computed on untokenized references (e.g., using SacreBLEU), not tokenized references. Another metric like BLEU would be helpful.
>
> **[Response-3]:** We agree with this point and thank you for highlighting that it's not a reason to reject. To make our results comprehensive, we show the evaluation results in different metrics as well, e.g., Bleu, SacreBleu, ChrF++, and COMET. Please consult **Response-1 to Reviewer CxMx**. In short, the improvements are still evident (2+ points) for all of them. We will also include them in our final version.
>
> >[Question-1] How do these BLEU scores compare with bilingual MT system scores?
>
> **[Answer-1]:** Due to the limited time, it is hard for us to search for an optimal architecture for each language pair in our corpus. Instead, we show comparison within a limited setting. For the setting and results, please refer to **Response-B3 to Reviewer 4Nxw**. In short, 1) MMT shows better performance (however, we hesitate to make such a claim that MMT is better than bilingual MT because the backbone may not be optimal), 2) Our method is still able to bring some benefits in a bilingual setting.
>
>
> >[Question-2] You use MUSE for evaluating your word embeddings, but couldn't you construct a word similarity graph directly from MUSE?
>
> **[Answer-2]:** Good point. Using MUSE directly will make the architecture more aesthetic, and to be honest, we considered this point during this project as well. However, using MUSE to build graphs has some potential issues too: 1) MUSE has no frequency information. It will be hard to define the transfer ratio, resulting in losing control of the edge importance. 2) When a word in MUSE is split into multiple subwords, things get more complex.
>
> >[Question-3] Have you considered building graphs for other relationships besides synonymy? For example, if in French word A is a hyponym of word B, and in English word C is a hyponym of word D, and we know that word A and word C are translations of each other, could the GNN somehow infer that word B and word D are more likely to be translations of each other?
>
> **[Answer-3]:** This is an interesting point. However, we are somewhat hesitant about its viability for translation. This could be due to hyponyms occurring at different levels. E.g., A and C are “poodle” in French and English, while B and D mean “dog” and “animal” respectively. Here, B and D should not be considered as translations of each other.

---

### Official Review · Reviewer_CxMx · 2023-08-05

**Soundness:** 3

**Excitement:**

3: Ambivalent: It has merits (e.g., it reports state-of-the-art results, the idea is nice), but there are key weaknesses (e.g., it describes incremental work), and it can significantly benefit from another round of revision. However, I won't object to accepting it if my co-reviewers champion it.

**Missing References:**

GNN has been extensively used in neural machine translation. It would be good to at least mention a few:
- https://aclanthology.org/P19-1320.pdf
- https://aclanthology.org/N18-2078v2.pdf
- https://arxiv.org/pdf/2012.03477.pdf
- https://openreview.net/pdf?id=B1fA3oActQ
- https://arxiv.org/pdf/1704.04675.pdf

 A few other papers that explored incorporating word alignment information into multilingual models:
- https://aclanthology.org/2020.emnlp-main.210.pdf
- https://aclanthology.org/2021.acl-long.21.pdf

This paper be used as a motivation: https://arxiv.org/pdf/2201.00075.pdf

**Paper Topic And Main Contributions:**

This paper aims to improve multilingual neural machine translation systems by explicitly increasing the representation similarity of lexical translations, even in cases where different writing systems are used. To do this, the paper designed a word embedding re-parametrization mechanism based on the graph neural network (GNN). The high-level idea is to inject the lexical translation information obtained from a word aligner in the form of a word translation graph, thus allowing message-passing across lexical translations.

The method is evaluated on two different datasets, one being the IWSLT2014 multilingual benchmark also used by some prior work (e.g. LASS by Lin et al. (2021) as cited in the paper), the other being a new WMT-data-based benchmark created by the authors, called WMT30. Results indicate that the proposed method improves the model performance by +2.3 BLEU on average for IWSLT and +1.1 BLEU on WMT30. Further analysis also showed that the graph-based embeddings perform better by word similarity on the MUSE dictionary.

**Questions For The Authors:**

A. I'm not entirely clear how is the graph transfer ratio $g_{i,j}^{s,t}$ (defined in L191) is used subsequently in Section 3.2? Is it used as initialization of the edge weights?
B. L421-422: What is the "original" embedding referring to here?

**Reasons To Accept:**

1. The research question addressed by this paper is a valid and important one in the context of multilingual translation models.
2. All experiments showed improvements over the baseline, both in terms of translation quality and word similarity benchmark.
3. The presentation of the paper is mostly clear and easy to follow.

**Reasons To Reject:**

1. My biggest concern of the paper is its evaluation. (1) The paper only reported tokenized BLEU, which has been shown in Post (2018) (https://aclanthology.org/W18-6319.pdf) to be very hard to reproduce. Plus, the findings of WMT 2022 metrics shared task (https://www.statmt.org/wmt22/pdf/2022.wmt-1.2.pdf) also suggested against using BLEU as the only metric for evaluation.

I know the authors probably chose to use the tokenized BLEU in order to keep consistent with Lin et al. (2021), which also reported tokenized BLEU. Yet, I would still strongly recommend using sacreBLEU and also report at least one of BLEURT or COMET. If it is hard to get those numbers for the models in Lin et al. (2021), I would rather drop that baseline.

2. While I understand the importance of the research question, I don't fully agree with the necessity or the benefit to use GNN for that. To me, there are simpler ways to inject such information, such as random substitutions (https://aclanthology.org/2020.emnlp-main.210.pdf) or contrastive learning (https://aclanthology.org/2021.acl-long.21.pdf). Have the authors considered those alternatives? Has there been any effort to verify that GNN is the better method?

3. Lots of important related work is omitted, which I'll elaborate on in the missing references section.

**Reproducibility:**

4: Could mostly reproduce the results, but there may be some variation because of sample variance or minor variations in their interpretation of the protocol or method.

**Reviewer Confidence:**

4: Quite sure. I tried to check the important points carefully. It's unlikely, though conceivable, that I missed something that should affect my ratings.

**Typos Grammar Style And Presentation Improvements:**

- The word "station" in Figure 1 appears quite arbitrary to me. I'm not sure what point that part of the figure is trying to make.
- L247: $\alpha_i$ is used without introduction
- L334: missing period
- Table 2: I would suggest (1) adding M2M/NLLB model as a reference (you don't have to be better -- just to give the readers an idea of where your model stands) (2) including a breakdown of comparing results of non-latin-script languages -- this will reinforce your earlier point that this would enable lexical knowledge sharing across writing systems.

---

> ### Author Rebuttal · Authors · 2023-08-28
>
> Thanks for your efforts on our paper. We summarize and answer your questions as follows:
>
> > [Reason to Reject-1]: My biggest concern about the paper is its evaluation: the paper only reported tokenized BLEU.
>
> **[Response-1]:** We agree with this point and release the evaluation results in other metrics (SacreBleu, Chrf++, and COMET) for the IWSLT14 dataset here.
>
> | SacreBLEU  | en-de | en-es | en-fa | en-ar | en-he | en-nl | en-pl | en-it |
> |------------|------:|------:|------:|------:|------:|------:|------:|------:|
> | baseline   |  27.6 |  35.1 |  16.8 |  19.5 |  27.2 |  30.8 |  17.1 |  29.0 |
> | GM-1hop    |  29.7 |  36.9 |  18.7 |  21.3 |  29.4 |  32.9 |  18.5 |  30.6 |
> | GM-2hop    |  29.7 |  37.1 |  18.7 |  21.1 |  29.1 |  33.0 |  18.5 |  30.8 |
> | GM-3hop    |  30.2 |  37.7 |  19.5 |  21.9 |  29.5 |  33.6 |  19.0 |  31.5 |
> | 3-hop Gain |   +2.6 |  +2.6 |  +2.7 |   +2.4 |   +2.3 |   +2.8 |   +1.9 |   +2.5 |
>
> | ChrF++     | en-de | en-es | en-fa | en-ar | en-he | en-nl | en-pl | en-it |
> |------------|------:|------:|------:|------:|------:|------:|------:|------:|
> | baseline      |  52.1 |  58.0 |  40.1 |  43.7 |  50.5 |  54.5 |  42.1 |  52.8 |
> | GM-1hop    |  54.0 |  60.0 |  41.8 |  45.4 |  52.5 |  56.5 |  43.6 |  54.6 |
> | GM-2hop    |  54.0 |  60.0 |  41.8 |  45.4 |  52.5 |  56.5 |  43.6 |  54.6 |
> | GM-3hop    |  54.4 |  60.3 |  42.2 |  45.9 |  52.9 |  56.8 |  43.9 |  55.0 |
> | 3-hop Gain |   +2.3  |   +2.3   |   +2.1 |   +2.2   |   +2.4 |   +2.3   |   +1.8  |   +2.2 |
>
> | Comet      | en-de | en-es | en-fa | en-ar | en-he | en-nl | en-pl | en-it |
> |------------|------:|------:|------:|------:|------:|------:|------:|------:|
> | baseline   |  75.5 |  79.8 |  74.2 |  76.2 |  78.8 |  78.7 |  72.9 |  78.3 |
> | GM-1hop    |  77.5 |  81.4 |  76.2 |  78.2 |  80.7 |  80.8 |  74.8 |  80.0 |
> | GM-2hop    |  77.5 |  81.7 |  76.1 |  78.3 |  80.9 |  80.9 |  74.8 |  80.2 |
> | GM-3hop    |  77.8 |  81.9 |  76.6 |  78.5 |  81.2 |  81.3 |  75.2 |  80.5 |
> | 3-hop Gain |   +2.3 |   +2.1 |   +2.4 |   +2.3 |   +2.4 |   +2.6 |   +2.3 |   +2.2 |
>
> **It shows that average of 2+ points improvement over there for all of the metrics**. This shows that our improvements remain consistent across a large spectrum of evaluation metrics.
>
> We also release the result on WMT30 here:
>
> | SacreBLEU  | EN-X | X-EN | EN-X | X-EN | EN-X | X-EN |
> |------------|:----:|:----:|:----:|:----:|:----:|:----:|
> | baseline   | 28.4 | 31.1 | 29.8 | 31.2 | 19.4 | 25.5 |
> | GM-1hop    | 29.1 | 31.6 | 30.4 | 31.5 | 20.0 | 26.7 |
> | GM-2hop    | 29.3 | 31.9 | 30.8 | 31.7 | 20.3 | 27.1 |
> | GM-3hop    | 29.2 | 31.6 | 30.7 | 31.7 | 20.4 | 27.3 |
> | 2-hop Gain |  +0.9 |  +0.8 |  +1.0 |  +0.5 |  +0.9 |  +1.6 |
>
> | ChrF++     | EN-X | X-EN | EN-X | X-EN | EN-X | X-EN |
> |------------|:----:|:----:|:----:|:----:|:----:|:----:|
> | baseline   | 52.5 | 57.1 | 53.9 | 56.4 | 42.8 | 49.7 |
> | GM-1hop    | 53.0 | 57.5 | 54.4 | 56.6 | 43.5 | 50.6 |
> | GM-2hop    | 53.3 | 57.6 | 54.8 | 56.6 | 43.8 | 50.9 |
> | GM-3hop    | 53.2 | 57.5 | 54.7 | 56.6 | 43.8 | 51.0 |
> | 2-hop Gain |  +0.8 |  +0.5 |  +0.9 |  +0.2 |  +1.0 |  +1.2 |
>
> | COMET      | EN-X | X-EN | EN-X | X-EN | EN-X | X-EN |
> |------------|:----:|:----:|:----:|:----:|:----:|:----:|
> | baseline   | 82.0 | 83.4 | 80.6 | 79.7 | 73.3 | 72.7 |
> | GM-1hop    | 82.8 | 83.9 | 81.2 | 80.2 | 74.2 | 74.0 |
> | GM-2hop    | 83.1 | 84.1 | 81.5 | 80.2 | 74.7 | 74.3 |
> | GM-3hop    | 83.0 | 83.9 | 81.5 | 80.1 | 74.8 | 74.5 |
> | 2-hop Gain |  +1.1 |  +0.7 |  +0.9 |  +0.5 |  +1.4 |  +1.6 |
>
> These results will also be included in the final version.
>
> > [Reason to Reject-2]: Tokenize BLEU is hard to reproduce. The comparison with Lin et al. (2021) is not suitable.
>
> **[Response-2]:** Firstly, we acknowledge that SacreBleu is great since it unifies measurements. However, here we use the same tokenizer Lin et al. (2021) used in their open-source scripts, therefore the results are already comparable. Secondly, we agree that there is no need to compare with particular scores of previous work as we are not doing competitions, especially when the motivations are different. We list their results here just to show that our reproduced baseline is fine.
>
> >[Reason to Reject-3]: While I understand the importance of the research question, I don't fully agree with the necessity or the benefit of using GNN for that. To me, there are simpler ways to inject such information, such as random substitutions (https://aclanthology.org/2020.emnlp-main.210.pdf) or contrastive learning (https://aclanthology.org/2021.acl-long.21.pdf).
>
> **[Response-3]:** These two papers are somewhat relevant, but we should mention that 1) they both introduce external data in the data augmentation or pre-training phase and 2) they both introduce an external tool like MUSE or a method like contrastive learning. It's hard to say whether they are "simpler" or not. Meanwhile, we don't think a mature tool, like GNN, will make the system that complex. However, we agree to include them in our related work in our final version. Thanks for your suggestion.
>
>
> > [Question For the Authors]: How is $g_{i,j}^{s,t}$ used in section 3.2? Is it used as initialization of the edge weights? What is the "original" embedding referring to here?
>
> **[Answers]:** We answer your questions one by one as follows:
> 1) As we mentioned in Lines 200-204: "The corresponding bilingual equivalence graph $G_{s,t}$ can be induced by filling an adjacency matrix using $g_{i,j}^{s,t}$. By element-wise summation of multiple $G_{s,t}$, we can merge multiple bilingual graphs into a single multilingual graph $G$". **Here, $G$ is the graph we used in section 3.2**.
> 2) It is used to build the weights of edges in the graph. Given you mentioned "initialization" here, in case of potential misunderstanding of the training fashion, we want to highlight that: **the graph is fixed during training**. The transfer pathways and ratio defined in the graph are the priors we want to inject into the embedding table. E.g., “bike” and “fiets” can transfer information with each other hence there is an edge with a weight between them in the graph, resulting in that even though they write in different ways, knowledge can explicitly transfer.
> 3) As a re-parameterized method, **note that there are two embedding tables here.** One is original and the other is re-parameterized. The former does not transfer information with GNN, and the latter is the one fed into the transformer. The two embedding tables are trained in an end-to-end way.
>
> Please refer to our responses to other Reviewers (E.g., Review 4Nxw) for a better understanding of the formalization.
>
> > [Missing Reference]: GNN has been extensively used in neural machine translation. It would be good to at least mention a few.
>
> Due to page limitations, we could only focus on the most relevant related approaches, but we are happy to include additional references in the camera-ready version. However, we should also mention that while the references provided by the reviewer are related, none of them actually focus on applying GNNs in the context of multilingual machine translation, and the motivation differs a lot as well. E.g., document translation [xu-etal-2021-document-graph] and introducing syntax in bilingual translation [bastings-etal-2017-graph,marcheggiani-etal-2018-exploiting] or embeddings [vashishth-etal-2019-incorporating].
>
>
> ---
>
> Lastly, given the fact that your biggest concern is “BLEU is the only metric”, we tried our best to make it comprehensive, where the main widely-used metrics are all included. Also, as reviewer-hsEL mentioned, **"using only one metric is weak but that is not a reason to reject it."**

---

### Official Review · Reviewer_4Nxw · 2023-08-09

**Soundness:** 4

**Excitement:**

4: Strong: This paper deepens the understanding of some phenomenon or lowers the barriers to an existing research direction.

**Paper Topic And Main Contributions:**

This paper proposes to re-parametrize the embedding layer of an MT system such that the embedding of a word in a given language is pulled towards that of its translations in the other language. Their graph-based approach relies on the probability of two words being aligned in the dataset. Extensive experiments and qualitative analyses demonstrate the effectiveness of the approach across two datasets.

**Questions For The Authors:**

I'm moving some suggestions for further experiments in this section, since they do not constitute grounds for rejection and are merely suggestions.

Regarding **B1/**, I would have found it useful to have two other baselines along the Weighted Sum:
1. One where the re-parametrization only relies on the random walk, with no extra learnable parameters; e.g., $E' = \left(\prod_1^k G\right) E$
2. One where the reparametrization only relies on extra learnable parameters, e.g $E_k = \rho(W_k E_{k-1} + b_k)$ and $E_0 = E$

On a related note, remark that the Weighted Sum baseline is defined as $GE$, instead of $E + GE$ as is done for the $k$-hop variants.This entails that the embedding for e.g. _bike_ is re-parametrized as a weighted sum of the wordpieces it is aligned with; crucially, these might not contain the wordpiece _bike_ itself.


Regarding **B2/**: To provide some concrete actionable feedback, this is something that could have been effectively tested by testing two other formulations of $G$, e.g. in the ablation experiments in §4.4:
1. A variant $G_\mathrm{equiv.}$ where $G$ is strictly controlled to only contain edges between known equivalent words (e.g., using MUSE dictionaries or wordnet lemmas)
2. A variant $G_\mathrm{transl.}$  where $G_{ij}$ is defined simply as the empirical probability of wordpiece $w_j$ occurring in the translation of a sentence containing the wordpiece $w_i$ (regardless of eflomal alignments).

In practice $G_\mathrm{equiv.}$ might not outperform $G$ owing to the limited number of edges that such an approach could yield; so such a comparison might call for controlling for the number of edges by pruning the least-weighted edges of $G$; nonetheless, if the argument of the authors stands, then we would expect performances using $G_\mathrm{equiv.}$ to be better than using $G_\mathrm{transl.}$.

**Reasons To Accept:**

- The approach is well thought out; on a personal note, I can also say it is likely to be useful in my future work
- The experiments demonstrate the effectiveness and efficiency of the approach, with a clear improvement in BLEU scores and a low memory footprint.

**Reasons To Reject:**

**A/** there a few methodological points that could be better handled. In particular:

**A1/** the authors only report 1 seed (as far as I can tell)

**A2/** the baselines do not include comparable methods, and in particular other embedding alignment approaches; e.g., Aji et al (2020) (cited by the authors)

**B/** While the method is proven to be effective, I am not completely sold on the narrative around it; i.e., that it is effective for the reasons suggested by the authors. This boils down to a few points of comparison that would be valuable, but are regrettably missing.

**B1/** One question that specifically comes to mind is that of the relative importance of the extra embedding weights, vs. that of the implicit random walk in the alignment graph. As it currently stands, the authors do not disentangle these two factors. Throughout the paper, the authors seem to attribute the success of their approach to the former, but do not control for the latter (though in fairness they do remark that the added parameters correspond to a fraction (2%) of the total embeddings).

**B2/** Another area where I'm uncertain what to make of the authors' claim concerns the question of the semantic similarity of items with high weighted edges in $G$. As the authors remark (line 195 sq.):
> [The graph definition] _is based on the intuition that when a pair of aligned words frequently co-occur, they 1) have higher confidence as equivalent words, and 2) the knowledge sharing between these two will benefit more context during training._

 As such the graph $G$ need not only describe equivalent word classes. Or put another way: it is plausible that the improvements that this method obtains are mostly due to point 2) in the citation above, rather than point 1).

**B3/** Lastly, it would have been interesting to verify the usefulness of the proposal on bilingual models., as such models constitute a useful point of reference. Reading from line 110&mdash;111:
> _our method adapts to massive language pairs and a large vocabulary._

It is unclear to me (at least unproven) that the method actually fares well on smaller settings.

**Edit after rebuttal:** The authors' rebuttal provided a supplementary set of results that alleviates my concerns as per this last point.

**Reproducibility:**

4: Could mostly reproduce the results, but there may be some variation because of sample variance or minor variations in their interpretation of the protocol or method.

**Reviewer Confidence:**

3: Pretty sure, but there's a chance I missed something. Although I have a good feel for this area in general, I did not carefully check the paper's details, e.g., the math, experimental design, or novelty.

---

> ### Author Rebuttal · Authors · 2023-08-28
>
> Thank you for your acknowledgment of our efforts and your in-depth comments, especially in part B. Also, we'd love to hear that it might be useful for your future work. We summarize and answer the questions as follows:
>
> > [Reason to Reject-A1] The authors only report 1 seed (as far as I can tell).
>
> **[Response-A1]:** Due to the limitation of resources, we use a fixed random seed for all of the experiments. However, we agree that multiple runs and significance analysis are great. We will try to include them in our final version.
>
> > [Reason to Reject-A2] The baselines do not include comparable methods, and in particular other embedding alignment approaches; e.g., Aji et al (2020) (cited by the authors)
>
> **[Response-A2]:** We agree that inducing baselines sharing the same motivation is great, however, there is no similar research in multilingual MT scenarios as far as we know. Sun et al. (2022) might be a case, however, as we mentioned in Lines 145-146 and Lines 134-138, the limitations of Romanization and transliteration still hold. We should also mention that Aji et al (2020) is designed for bilingual MT and it relies on the pretraining method, which is somewhat hard to compare with.
>
> > [Reason to Reject-B1] One question that specifically comes to mind is that of the relative importance of the extra embedding weights, vs. that of the implicit random walk in the alignment graph. As it currently stands, the authors do not disentangle these two factors... I would have found it useful to have two other baselines along the Weighted Sum:
> > 1) One where the re-parametrization only relies on the random walk, with no extra learnable parameters; e.g., $E^{k}=(\prod_{i=1}^{k}G)E$
> > 2) One where the reparametrization only relies on extra learnable parameters, e.g., $E_k=\rho(W_k E_{k-1} + b_k)$
>
> **[Response-B1]:** This is a really insightful point. We appreciate your interest in addressing these two factors, especially for the random walk setting. It is also efficient since we can calculate $G^*=\prod_{i=1}^{k} G$ before training. We will try to introduce this part in our final version. For the second point, we are hesitating to claim any potential performance change can be solely attributed to the extra parameter. Because the extra parameters are introduced by the projection function, which works as a part of GNN to adjust aggregation. A separate form, e.g., the one you mentioned in the second part, would lose this capability directly.
>
>
> > [Reason to Reject-B2]:  Another area where I'm uncertain what to make of the authors' claim concerns the question of the semantic similarity of items with high-weighted edges in $G$. As the authors remark (line 195 sq.): [The graph definition] is based on the intuition that when a pair of aligned words frequently co-occur, they 1) have higher confidence as equivalent words, and 2) the knowledge sharing between these two will benefit more context during training.
>
> **[Response-B2]:** We deeply value and appreciate your efforts on the details of our paper. We try to reply to you in the following aspects:
>
> 1) Firstly, we wanna explain the second point of intuition a little bit, i.e., _"the knowledge sharing between these two will benefit more context during training."_ Suppose a verb A with the meaning of 'bike/cycle to somewhere' in one language. It could be translated to 'bike', 'cycle', and 'pedal' in English, and the translation of 'bike' and 'cycle' occurs a lot of time while only a few times for 'pedal'. Intuitively, the weight between A and 'pedal' should be low since its influence is low as well. We will modify this point in our final version in case any misleading. **In the following responses, we solely focus on the potential or issue with the design of $G_{equiv}$ and $G_{transl}$ you mentioned.**
> 2) Secondly, we appreciate the idea of utilizing MUSE to construct the graph $G_{equiv}$, as also suggested by Reviewer hsEL (albeit from an aesthetic perspective rather than an ablation one). However, it's important to note that this approach may introduce certain potential issues.  E.g., frequency information will be lost in this case, and the thing gets more complex when words in MUSE are split into subwords.
> 3) Lastly, your design of $G_{transl}$ is really an interesting point, which makes sense to some extent and gets rid of using an external tool, e.g., alignment or MUSE. One potential issue would be like: it will make the graph dense since the edge now is defined upon the co-occurrence of any word pairs within sentence pairs. Maybe defining the weights using pointwise mutual information and using a threshold to prune part of them is a way. We will try to explore it in the future.
>
>
> > [Reason to Reject-B3]: Lastly, it would have been interesting to verify the usefulness of the proposal on bilingual models. It is unclear to me (at least unproven) that the method actually fares well in smaller settings.
>
> **[Response-B3]:**  Yes, it is an interesting point. We are also curious about this. Intuitively, our method might also work for bilingual translation. We conducted additional experiments to check this point:
>
> Due to the limitation of time, we randomly picked a medium-resource and a low-resource language pair from the WMT30 dataset: EN-DA (1M) and EN-FA (100K), to conduct the experiment individually (Note that the graph is rebuilt for each as well). Unfortunately, we can't search for an optimal backbone in a short time, hence we use a rule of thumb setting, i.e., transformer-base (6 layers, 512 embeddings dim, 1024 feedforward, 4 attention heads). We shrink the vocabulary size to 16K and the batch size to 80K and 40K for medium- and low-resource respectively. We compare the baseline with GraphMerge-3hop setting as follows:
>
> |                | EN-DA | DA-EN | EN-AF | AF-EN |
> |----------------|-------|-------|-------|-------|
> | Bilingual Baseline       | 35.3  | 35.4  | 31.7  | 35.9  |
> | Bilingual GraphMerge-3hop| 35.6  | 35.8  | 33.9  | 39.6  |
> | WMT30 Baseline            | 36.9  | 39.7  | 38.4  | 48.2  |
> | WMT30 GraphMerge-3hop     | 38.1  | 40.0  | 38.5  | 50.1  |
>
> It shows that in an "extreme" setting (bilingual setting), our method still brings some benefits, especially on the low-resource part. We also list the results of our MMT trained on WMT30 in these directions for your reference. It is worth noting that, although MMT seems to be better here, we are hesitating to claim it, since the backbone of bilingual translation may not be optimal (e.g., model size).

---

### Official Review · Reviewer_8KXe · 2023-08-11

**Soundness:** 4

**Excitement:**

4: Strong: This paper deepens the understanding of some phenomenon or lowers the barriers to an existing research direction.

**Missing References:**

[imanigooghari-etal-2021-graph] and [imani-etal-2022-graph] show methods to improve word alignment quality.


**Paper Topic And Main Contributions:**

This paper presents a method to decrease the embedding distance of similar words and improve the machine translation quality in an end-to-end model.
The proposed framework, first extracts lists of word pairs with similar meanings using word alignments.
A graph network transforms the original embeddings to a new space where similar words are better aligned across languages.
An encoder-decoder transformer model then uses the new embeddings for translation, and the whole model is trained end-to-end.

The experiments include the evaluation of two datasets containing several language pairs ranging from low-resource to high-resource. The results show consistent improvements over baseline when using more graph layer hops.
The paper also includes an ablation study over the quality of word alignments and tying encoder and decoder embeddings, as well as other experiments to study the quality of the embeddings and the performance of the proposed model.



**Questions For The Authors:**

Question A: Your framework only uses old word aligners. There are several established word aligners such as SimAlign and AwesomeAlign that perform much better. Is there a reason that you didn't use them? Because I think it would further improve your performance.

Question B: With the current model, the rare words that don't occur in the word alignments won't benefit from this model at all. Do you have any plans or solutions for them?


**Reasons To Accept:**

I like this paper and I like to see it accepted at this conference.
- Although using GNNs over word alignments to improve the quality of representations has been proposed before (imani-etal-2022-graph), using it in an end-to-end fashion in an MNMT model is novel and could be used more in the community.
- The experiments include several language pairs and show the generalizability of the proposed model. I like the fact that Table 2 has different sections for high, medium, and low resource performance.
- Including the ablation studies and the additional analysis makes this work more valuable.
- The authors include as many technical details as possible that facilitate to reproduce the results. They also claim to release the codes upon acceptance of the paper.


**Reasons To Reject:**

- Table 4, 5: It is established to use Precision@k (commonly 1) to check the quality of Bilingual Lexicon Induction.
Using the MUSE dataset with other metrics could be misleading.
Although we know the quality of translations has improved, it could be the case that the nearest neighbors of each word could change during the graph transformation. Looking at average similarity is not a good metric to check this.

- Section 5.2: Adding more pairs that are extracted from word alignments made these results questionable.
In a way, the word alignment model might have a bias on the graph model to make specific types of words closer to each other (for example only nouns). Testing the outputs with datasets extracted from the same word aligners is not valid.

- The experiments don't include baselines that use multilingual language models as encoders. It might be the case that a multilingual LM that only uses monolingual data performs better than the proposed method.



**Reproducibility:**

5: Could easily reproduce the results.

**Reviewer Confidence:**

4: Quite sure. I tried to check the important points carefully. It's unlikely, though conceivable, that I missed something that should affect my ratings.

---

> ### Author Rebuttal · Authors · 2023-08-28
>
> Thank you for carefully reading our paper and insightful suggestions. We’re delighted that you pointed out the connections between our paper and [imanigooghari-etal-2021-graph, imani-etal-2022-graph]. We agree on this point and will put them in the related work in our final version. We summarize and answer your questions as follows:
>
> > [Reason to Reject-1]: Why not use Precision@k (commonly 1) to check the quality of Bilingual Lexicon Induction instead of representational similarity?
>
> **[Response-1]:** One reason is that we intend for the analysis to align closely with our motivation, namely, the enhancement of representational word similarity. Additionally, relying solely on high-level Precision@k scores in the context of BLI might not necessarily ensure that the representation distances are close. To illustrate this, consider an extreme scenario where pairs of word embeddings are meticulously arranged along two parallel lines in the space. In such a case, the distance between the parallel lines does not impact the BLI outcome.
>
> > [Reason to Reject-2]: Section 5.2: Adding more pairs that are extracted from word alignments made these results questionable. In a way, the word alignment model might have a bias on the graph model to make specific types of words closer to each other (for example only nouns). Testing the outputs with datasets extracted from the same word aligners is not valid.
>
> **[Response-2]:** We think there might be a misunderstanding here. In our case, the word pairs for testing are still sourced from MUSE, rather than alignments. However, we extended the MUSE to non-English pairs [refer to Lines 488-493]. However, the graph structure remains unchanged, still exclusively based on English-centric alignments. Our analysis shows that our method can also enhance the representational similarities among non-English word pairs. This enhancement is achieved as the pivot (English) facilitates the transfer of knowledge through our multi-hop mechanism.
>
>
> >[Reason to Reject-3]: The experiments don't include baselines that use multilingual language models as encoders. It might be the case that a multilingual LM as the encoder that only uses monolingual data performs better than the proposed method.
>
> **[Response-3]:** Using multilingual LM as an encoder is an interesting setting. However, some issues would arise: 1) As a kind of pre-train + finetune fashion, there is no guarantee that the knowledge learned from the multilingual LM (like multilinguality of embeddings) will not be washed out after finetuning [liu-etal-2020-multilingual-denoising], resulting in situations where pretraining won't work well for high-resource language directions (even hurt the performance sometimes). 2) The model size is bundled with the pretraining model as well. 3) Lastly, we should also mention that the main issue we target, i.e., the lack of explicit knowledge transfer among words in different scripts, still holds for multilingual LMs.
>
>
> > [Question-1]: Your framework only uses old word aligners. There are several established word aligners such as SimAlign and AwesomeAlign that perform much better. Is there a reason that you didn't use them? Because I think it would further improve your performance.
>
> **[Answer-1]:** We totally agree that better alignment quality could further improve the performance. This point has been demonstrated already as our ablation study shows that using Eflomal is slightly better than using FastAlign (see Table 3, page 6). However, we also agree that taking a look at how far our architecture can go is valuable. We will try to include them in our next version.
>
> > [Question-2]: With the current model, the rare words that don't occur in the word alignments won't benefit from this model at all. Do you have any plans or solutions for them?
>
> **[Answer-2]:** This is indeed a good question and we also considered this point throughout the project. 1) Firstly, we should note that "rare words" are a general problem in most NLP scenarios, not only a particular flaw in our architecture. 2) Secondly, our method (although not mentioned in the paper) may have the potential to alleviate this issue by manually adding the edges between rare words with those counterparts across languages. 3) Also, relying on technologies in cross-lingual word embeddings and then conducting BLI (or other kinds of retrieval) across languages to build graphs, may have the potential to cover more links among rare words. However, other designs need to be considered since the weights of the graph become hard to define in this case.

---

### Meta-Review · Area_Chair_5PxZ · 2023-09-08

**Recommendation:** 5

**Metareview:**

Tokenization methods in SoTA MT consider subwords with the same orthography as equivalent, which does not make much sense with totally unrelated languages. The paper proposes using semantic equivalence instead of orthographic equivalence. They achieve systematic improvements in translation quality in all major automatic metrics (some of which were added during the discussion period) both in small (IWSLT14) and large (WMT) data setup.

The paper received 4 reviews, 3 of which are very positive about the paper, both in terms of soundness and excitement, one review was very negative. Two of the reviewers raised important issues concerning the evaluation metrics, which were later clarified during the discussion period and the only negative-leaning reviewer increased their score.

---

### Decision · Program_Chairs · 2023-10-07

**Decision:**

Accept-Main

**Comment:**

Tokenization methods in SoTA MT consider subwords with the same orthography as equivalent, which does not make much sense with totally unrelated languages. The paper proposes using semantic equivalence instead of orthographic equivalence. They achieve systematic improvements in translation quality in all major automatic metrics (some of which were added during the discussion period) both in small (IWSLT14) and large (WMT) data setup.

The paper received 4 reviews, 3 of which are very positive about the paper, both in terms of soundness and excitement, one review was very negative. Two of the reviewers raised important issues concerning the evaluation metrics, which were later clarified during the discussion period and the only negative-leaning reviewer increased their score.